# META-GAN FOR FEW-SHOT IMAGE GENERATION

**Arvind Sridhar**
Department of Computer Science
Stanford University, Stanford, CA
`asridhar@cs.stanford.edu`

## ABSTRACT

While Generative Adversarial Networks (GANs) have rapidly advanced the state of the art in deep generative modeling, they require a large amount of diverse datapoints to adequately train, limiting their potential in domains where data is constrained. In this study, we explore the potential of few-shot image generation, enabling GANs to rapidly adapt to a small support set of datapoints from an unseen target domain and generate novel, high-quality examples from that domain. To do so, we adapt two common meta-learning algorithms from few-shot classification–Model-Agnostic Meta-Learning (MAML) and Reptile–to GANs, meta-training the generator and discriminator to learn an optimal weight initialization such that fine-tuning on a new task is rapid. Empirically, we demonstrate how our MAML and Reptile meta-learning algorithms, meta-trained on tasks from the MNIST and SVHN datasets, rapidly adapt at test time to unseen tasks and generate high-quality, photorealistic samples from these domains given only tens of support examples. In fact, we show that the generated image quality of these few-shot adapted models is on par with that of a baseline model vanilla-trained on thousands of samples from the same domain. Intriguingly, meta-training also takes substantially less time to converge compared to baseline training, indicating the power and efficiency of our approach. We also demonstrate the generalizability of our algorithms, working with both CNN- and Transformer-parametrized GANs. Overall, we present our MAML and Reptile meta-learning algorithms as effective strategies to enable few-shot image generation, improving the feasibility of deep generative models in practice.

## 1 INTRODUCTION

With the rapid adoption of deep generative models to model complex visual data distributions, generating novel high-quality images within a certain domain has become a tractable task. Generative Adversarial Networks (GANs) (Goodfellow et al., 2014) have become the state of the art in deep generative modeling, with the best GANs generating novel images that well-preserve the style and content fidelity of the training domain and scaling up seamlessly to large datasets (hundreds by hundreds of pixels) (Karras et al., 2019; Miyato et al., 2018; Karras et al., 2020; Brock et al., 2019).

However, modern GAN architectures require extremely large datasets to effectively learn the training distribution, limiting their potential and practicality for domains with scarce data (Ojha et al., 2021). By contrast, humans are excellent *few-shot learners*, able to generate high-quality visual samples within a certain target domain after observing just a few training examples. To overcome these data constraints, several works propose transfer learning from a large source domain (Wang et al., 2018; 2020), yet even these techniques overfit when the target has only a few samples (Li et al., 2020).

On the other hand, few-shot image recognition has made great strides towards improving classifiers' ability to quickly adapt to sparse target domains. Model-Agnostic Meta-Learning (MAML) has become the state of the art for $k$-shot learning with $k$ as small as 1, guiding the model to rapidly learn robust feature representations that well-separate the $n$-way data distribution. MAML meta-trains the classifier such that, with just a few fine-tuning steps on a sparse target domain, it can effectively learn the class decision boundaries (Finn et al., 2017). Building upon this paradigm, Nichol et al. (2018) developed Reptile, a first-order approximation of MAML that improves meta-training stability and efficiency, enabling faster convergence with empirically similar results on many few-shot benchmarks.

In this work, we apply the MAML and Reptile meta-training algorithms to deep generative modeling, optimizing a GAN such that, at test time, the generator and discriminator can rapidly fine-tune to a previously-unseen target domain with only tens of training examples. We experiment with both CNN- and Transformer-parametrized GANs. Recently, Transformers (Vaswani et al., 2017) have gained much attention for their ability to effectively solve a broad and diverse array of challenges in machine learning. Specifically, Vision Transformers (ViTs) have been able to achieve state-of-the-art results in image classification with only relatively few gradient steps fine-tuning on different target domains (Dosovitskiy et al., 2021). As such, Transformers are a natural architecture to consider for few-shot image generation applications. Prior studies have developed and trained Transformer-based GANs (Lee et al., 2021; Jiang et al., 2021), achieving results that outperform the state-of-the-art CNN-based StyleGAN-V2 (Karras et al., 2020) on CIFAR-10 and CelebA. However, no studies to date have explored the few-shot adaptation performance of these models to sparse target domains.

In this work, we develop 2 novel meta-training algorithms for GANs based on the MAML and Reptile paradigms. Our goal is **few-shot image generation**: given a sparse target domain with only tens of support examples, our GAN is able to rapidly adapt to this domain's distribution (without overfitting) and generate novel, photorealistic examples on par with a standard GAN vanilla-trained on a large dataset from the same domain. Our algorithms are also **model-agnostic**, in that they can work with any type of generator or discriminator architecture (CNN- or Transformer-based). We demonstrate the efficacy of our meta-learning algorithms on both MNIST and SVHN digit generation benchmarks, with both CNN- and Transformer-based GANs. We meta-train on tasks of generating digits '0' to '8', and test on tasks of generating the digit '9', to accurately gauge the few-shot adaptation ability. Given test-time supports of only size 10, both MAML and Reptile succeed in generating images with comparable visual quality and fidelity (quantitatively measured by FID, KID, and IS scores) to the same GAN vanilla-trained on all training images of '9' (baseline). Intriguingly, both MAML and Reptile also take substantially ($5\times$) shorter to meta-train to convergence compared to the baseline.

Further, we find that Transformer-based GANs outperform CNN-based GANs across the board. This affirms our hypothesis that Transformers are better few-shot learners, though this outperformance could also be due to the fact that Transformers are simply better at image modeling than CNNs. Overall, we develop and showcase the power of two novel meta-learning algorithms–MAML and Reptile–for deep generative modeling, enabling diverse GAN architectures to meta-train efficiently, rapidly adapt to novel target domains, and generate novel high-quality images on par with the baseline.

## 2 PROBLEM STATEMENT AND SETUP

Concretely, we seek to adapt the Model-Agnostic Meta-Learning (MAML) and Reptile algorithms for GANs, meta-training them for rapid few-shot adaptation. We will focus on the MNIST (re-scaled to $32 \times 32$) and SVHN digit datasets. We define our two major paradigms of training below:

**Baseline Training**: train the generator and discriminator as normal using all images of the digit '9' in the training set, so the models fully learn the distribution of the nines to produce them at test time.

**Meta-Training**: train the generator and discriminator (using MAML/Reptile) on meta-tasks consisting only of the digits '0' to '8', so the task of generating nines remains unseen until test time.

At test time, the meta-trained model is forced to adapt to the unseen task of generating '9' given only a few $(1 - 20)$ support examples of '9' from the training set. The quality of both models' generated nines is then evaluated visually and quantitatively. For quantitative evaluation, we use the Frechet Inception Distance (FID), Kernel Inception Distance (KID), and Inception Score (IS) metrics, common in the GAN literature, comparing the generated nines to all the images of '9' in the test set.

We define a **meta-task** at both train and test time consisting of a single digit label $d$, $k_s$ total support examples of digit $d$ sampled from the training dataset, and $k_q$ total query examples of digit $d$ sampled from the test dataset. For the majority of our experiments, we fix $k_s = 10$ and $k_q$ to the total number of examples of digit $d$ in the test dataset (for comprehensive evaluation). Concretely, this means that at test time, the meta-trained model must learn to generate high-quality nines with only 10 support examples from the training set, while the baseline has trained on all the images of '9' in the train set.

Our meta-task setup differs from Liang et al. (2020) in one crucial way: we limit each task to examples of only a single digit ('0' to '8' at train time, '9' at test time), while Liang et al. (2020) allowed

meta-training tasks to have a mixture of samples of digits '0' to '8'. We believe that our method is the correct approach, as training on mixed tasks introduces a fundamental train-test mismatch. Concretely, meta-training and testing on single-digit tasks enforces task consistency: the model learns how to produce high-quality images of a single digit given a small support of that digit. Training on mixed tasks instead teaches the model how to model an entire distribution of digits in a few-shot manner, which is not the test-time objective. We find that, while Liang et al. (2020)'s method works fine on MNIST (generating results on par with baseline, as they show in their paper), it falls apart with the more complicated SVHN (unable to produce high-quality nines), manifesting these issues.

To parametrize our GANs, we utilize two state-of-the-art architectures: ResNet-parametrized SN-GAN (Res-SNGAN) (Miyato et al., 2018) and TransGAN (Jiang et al., 2021), with a pure Transformer architecture. We train Res-SNGAN from scratch and utilize TransGAN pre-trained on CIFAR-10 generation. For both models, we set our discriminator to only classify inputs as real or generated.

## 3    TECHNICAL APPROACH

### 3.1    META-TRAINING ALGORITHM

Our meta-training algorithm is formally outlined in Algorithm 1. We meta-train the generator and discriminator using stochastic gradient descent, one update per meta-training task sampled. We also progressively reduce the meta-learning rate as meta-training goes on, improving training stability.

### 3.2    INNER-LOOP OPTIMIZATION: MAML, REPTILE

For the inner loop optimization step above, we adapt the MAML and Reptile algorithms. Algorithm 2 details the inner loop algorithm for MAML, and Algorithm 3 for Reptile. In practice, we set $k_s = k_q = 10$, $R = 10000$, $r = 12$, $\alpha_{\text{inner}} = 0.0002$, process the entire support/query set as a single batch, optimize the generator loss once every 2 steps, and optimize the discriminator loss every step.

## 4    EXPERIMENTAL RESULTS

### 4.1    COMPARING TO BASELINE: MNIST

To begin, we compare the performance Res-SNGAN models trained with the baseline method and meta-training (MAML and Reptile) on MNIST, evaluating the visual quality of the generated images of nines as well as the FID, KID, and IS scores. The visual results are shown in Figure 1, where the meta-trained models are given 10 images of '9' and $r = 100$ fine-tuning iterations at test time. Despite these constraints, **both MAML and Reptile generate nines with quality on par with, if not better than, the baseline**, trained on the entire training distribution of nines (6000 images) for 100000 iterations. Table 1 presents the quantitative evaluation of MAML and Reptile relative to the baseline for both Res-SNGAN and TransGAN. Both MAML and Reptile achieve comparable FID and IS scores to the baseline for both architectures, and outperform the baseline on KID. A surprising result was respect to train time of meta-training vs baseline: we discuss this in detail in Appendix B.1.

Further, Table 1 compares the performance of Res-SNGAN and TransGAN. TransGAN baseline training outperforms that of Res-SNGAN, as expected (and following the literature), given that TransGAN is pre-trained on CIFAR-10 and is a more expressive model. MAML and Reptile show similar trends as Res-SNGAN, performing on par with the baseline (better in the case of KID). The relative improvement of KID over the baseline seems to be greater for TransGAN vs Res-SNGAN, though the gap in FID and IS seems to be slightly larger. This indicates that TransGAN does not provide any special boost to few-shot meta-learning, contrary to our original hypothesis. However, this could also indicate that the Transformer was just better at fitting the baseline nines distribution, and thus comparing MAML and Reptile relative to this is not a fair comparison of the two models.

### 4.2    COMPARING TO BASELINE: SVHN

We also compare the performance of our Res-SNGAN and TransGAN trained with the baseline, MAML, and Reptile on SVHN. The visual results are shown in Figure 2, exhibiting similar results

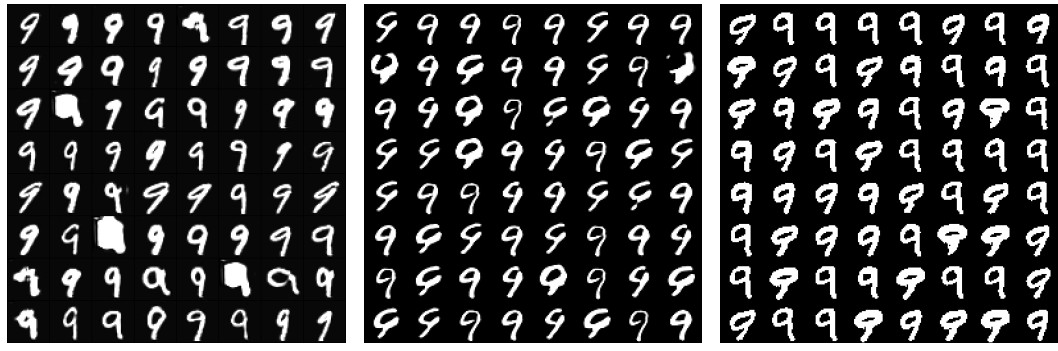

(a) Baseline Training          (b) MAML Meta-Training          (c) Reptile Meta-Training

Figure 1: Comparing test-time performance of Res-SNGAN models {meta-}trained on MNIST with various methods. The quality of the generated nines by meta-training (MAML/Reptile) is on par with baseline. Of the two meta-training approaches, Reptile seems to yield slightly better visual results.

| Training Method | Model | Total Train Time (s) | FID | KID | IS |
|---|---|---|---|---|---|
| Baseline | Res-SNGAN | 16567 | **33.59** | 0.0137 | **1.677** |
| MAML Meta-Train | Res-SNGAN | 6841 | 37.12 | 0.0138 | 1.435 |
| Reptile Meta-Train | Res-SNGAN | **4688** | 37.37 | **0.0136** | 1.402 |
| Baseline | TransGAN | 26641 | **30.13** | 0.0121 | **1.798** |
| MAML Meta-Train | TransGAN | 11784 | 34.17 | 0.0118 | 1.513 |
| Reptile Meta-Train | TransGAN | **8758** | 34.11 | **0.0117** | 1.497 |

Table 1: Comparing total training time and quantitative test-time performance of the baseline, MAML, and Reptile with both Res-SNGAN and TransGAN on MNIST. MAML and Reptile achieve comparable scores to the baseline, even achieving a lower KID score. This trend holds across model architectures. As expected, TransGAN outperforms Res-SNGAN, and surprisingly, meta-training both MAML and Reptile to convergence takes substantially shorter ($3\times$ to $4\times$) than baseline training.

as MNIST. Both MAML and Reptile are able to rapidly adapt to the distribution of nines with the given support and generate high-quality images of '9' on par with the baseline (trained on 7000 images of nines). Table 2 showcases a similar trend: the performance of MAML and Reptile w.r.t FID, KID, and IS is comparable to that of the baseline. However, the gap between the baseline and meta-training w.r.t. these metrics is slightly greater. Also, as with the MNIST, both MAML and Reptile take substantially shorter to meta-train to convergence than the baseline. Overall, the trends seen with SVHN mirror MNIST, reinforcing the generalizability of our meta-training algorithms.

TransGAN exhibits a similar trend as with MNIST, though the gap between meta-training and baseline performance is smaller than with Res-SNGAN. With the harder-to-model SVHN dataset, the Transformer seems to have provided meta-training with an advantage, improving few-shot adaptation.

## 5  CONCLUSIONS AND FUTURE DIRECTIONS

In this study, we develop two novel meta-learning algorithms–MAML and Reptile–for **few-shot image generation**, enabling GANs to rapidly adapt to a small support set from an unseen target domain. Empirically, our algorithms enable rapid 10-shot adaptation on both MNIST and SVHN, generating images with quality on par with a baseline trained exclusively on that task, in a fraction of the training time. MAML and Reptile generalize to both CNN- and Transformer-parametrized GANs.

We encourage future work to further investigate the ability of Transformers to improve few-shot adaptation performance relative to the baseline. In this work, we observe that Transformers offer an advantage over Res-SNGAN when the dataset becomes more complicated, i.e. SVHN. An interesting study would attempt to uncover the limit of this improvement, particularly evaluating the performance of TransGAN vs Res-SNGAN on larger datasets like CIFAR-10, LSUN, and perhaps even ImageNet.

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

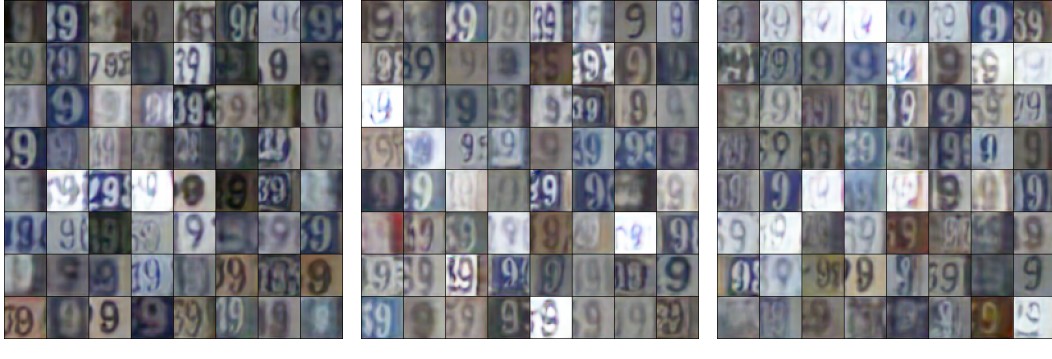

| (a) Baseline Training | (b) MAML Meta-Training | (c) Reptile Meta-Training |

Figure 2: Comparing test-time performance of Res-SNGAN models {meta-}trained on SVHN with various methods. Like MNIST, MAML and Reptile are meta-trained on tasks of digits '0' through '8' and, at test time, are only given 10 images of '9' to adapt. Also like MNIST, the baseline is vanilla-trained on all the '9's in the training set (about 7000 images). As can be seen, the quality of the generated nines by the meta-trained, few-shot adapted algorithms is on par with the baseline.

| Training Method | Model | Total Train Time (s) | FID | KID | IS |
|---|---|---|---|---|---|
| Baseline | Res-SNGAN | 29231 | **42.34** | **0.0412** | **2.378** |
| MAML Meta-Train | Res-SNGAN | 13949 | 47.11 | 0.0487 | 2.012 |
| Reptile Meta-Train | Res-SNGAN | **9328** | 46.34 | 0.0481 | 2.043 |
| Baseline | TransGAN | 41291 | **38.12** | **0.0397** | **2.617** |
| MAML Meta-Train | TransGAN | 18679 | 41.88 | 0.0453 | 2.298 |
| Reptile Meta-Train | TransGAN | **12397** | 41.47 | 0.0429 | 2.317 |

Table 2: Comparing the total training time and quantitative test-time performance of the baseline, MAML, and Reptile with both Res-SNGAN and TransGAN on SVHN. FID, KID, and IS scores are evaluated relative to all the images of '9' in the test set (about 2000 images). As can be seen, MAML and Reptile achieve comparable quantitative scores to the baseline, though the gap is slightly bigger than MNIST. This trend holds across model architectures, showcasing the generalizability of our method. Like MNIST, TransGAN outperforms Res-SNGAN across all metrics, and meta-training both MAML and Reptile to convergence takes substantially shorter (about $3\times$) than baseline training.

## A    META-TRAINING ALGORITHMS

We outline our overall meta-training framework in Alg. 1, MAML in Alg. 2, and Reptile in Alg. 3.

For both MAML and Reptile, we clone the meta-networks first in order to perform fine-tuning, run the inner loop optimization (i.e. run standard GAN training for $r$ iterations), and update the meta-networks with the results. For MAML, we utilize both the support and query sets (we set $k_s = k_q = 10$) of the task in order to compute the meta-gradients (i.e. the gradients of the meta-networks); this follows standard MAML, where the loss on the query set after fine-tuning is back-propagated through the inner loop (Finn et al., 2017). Because the generator loss is based on the output of the discriminator, we need to clone and fine-tune the discriminator twice, once on the support set (to calculate the discriminator's meta-gradient) and once on the query set (to calculate the generator's meta-gradient).

Reptile's inner loop is significantly simpler, as we only require the support set and not the query set to calculate the meta-gradients, following Nichol et al. (2018). As a result, we only clone and fine-tune the generator and discriminator once, and the first order approximation yields the meta-gradients as simply the difference between the original meta-parameters $\theta, \phi$ and the fine-tuned parameters $\hat{\theta}, \hat{\phi}$.

---

**Algorithm 1:** Meta-Training Algorithm for Fast GAN Adaptation

---

**Input** : Number of meta-training iterations $R$, inner loop learning rate $\alpha_{\text{inner}}$ and betas $\beta_{\text{inner}}$
**Output :** Meta-trained GAN $G_\theta(z)$, $D_\phi(x)$

1   Initialize $G_\theta(z)$, $D_\phi(x)$
2   **for** $i \leftarrow 1$ **to** $R$ **do**
3     |   $T(c, k_s, k_q) \leftarrow$ sample(training_set, test_set)     # sample meta-training task
4     |   $\alpha_{\text{meta}} \leftarrow 1 - i/R$     # update meta-learning rate
5     |   inner_optimizers $\leftarrow$ get_optim_adam($\alpha_{\text{inner}}, \beta_{\text{inner}}$)
6     |   **inner_loop_optimization**($G_\theta$, $D_\phi$, $T$, $\alpha_{\text{meta}}$, inner_optimizers)
7   **end**

---

**Algorithm 2:** Model-Agnostic Meta-Learning (MAML) for GAN Meta-Training

---

**Input** : Current generator $G_\theta(z)$, discriminator $D_\phi(x)$, task $T$, number of inner loop steps $r$, inner loop learning rate $\alpha_{\text{inner}}$, meta-learning rate $\alpha_{\text{meta}}$
**Output :** Updated generator and discriminator $G_\theta(z)$, $D_\phi(x)$

1   $G_{\hat{\theta}} \leftarrow G_\theta$     # clone generator
2   $D_{\hat{\phi}} \leftarrow D_\phi$     # clone discriminator
3   $x_{\text{supp}}, \ x_{\text{query}} \leftarrow T$
4   **for** $i = 1$ **to** $r$ **do**
5     |   Generate fake samples $\tilde{x} = G_{\hat{\theta}}(z)$
6     |   Run the discriminator $D_{\hat{\phi}}(x)$ on real $x_{\text{supp}}$ and fake $\tilde{x}$
7     |   Compute the discriminator loss, backpropagate and update $D_{\hat{\phi}}(x)$ using $\alpha_{\text{inner}}$
8   **end**
9   Compute the discriminator **meta-gradient**: $\nabla D = \nabla_\phi \mathcal{L}_{D,\hat{\phi}}(x_{\text{query}}, \tilde{x})$
10   $D_{\hat{\phi}} \leftarrow D_\phi$     # clone discriminator
11   **for** $i = 1$ **to** $r$ **do**
12     |   Generate fake samples $\tilde{x} = G_{\hat{\theta}}(z)$
13     |   Run the discriminator $D_{\hat{\phi}}(x)$ on real $x_{\text{query}}$ and fake $\tilde{x}$
14     |   Compute the discriminator loss, backpropagate and update $D_{\hat{\phi}}(x)$ using $\alpha_{\text{inner}}$
15     |   Compute the generator loss, backpropagate and update $G_{\hat{\theta}}(z)$ using $\alpha_{\text{inner}}$
16   **end**
17   Compute the generator **meta-gradient**: $\nabla G = \nabla_\theta \mathcal{L}_{G,\hat{\theta}}(\tilde{x})$
18   **Meta-update generator**: $\theta \leftarrow \theta - \alpha_{\text{meta}} \nabla G$
19   **Meta-update discriminator**: $\phi \leftarrow \phi - \alpha_{\text{meta}} \nabla D$

---

**Algorithm 3:** Reptile for GAN Meta-Training

---

**Input** : Current generator $G_\theta(z)$, discriminator $D_\phi(x)$, task $T$, number of inner loop steps $r$, inner loop learning rate $\alpha_{\text{inner}}$, meta-learning rate $\alpha_{\text{meta}}$
**Output :** Updated generator and discriminator $G_\theta(z)$, $D_\phi(x)$

1   $G_{\hat{\theta}} \leftarrow G_\theta$     # clone generator
2   $D_{\hat{\phi}} \leftarrow D_\phi$     # clone discriminator
3   $x_{\text{supp}}, \ x_{\text{query}} \leftarrow T$
4   **for** $i = 1$ **to** $r$ **do**
5     |   Generate fake samples $\tilde{x} = G_{\hat{\theta}}(z)$
6     |   Run the discriminator $D_{\hat{\phi}}(x)$ on real $x_{\text{supp}}$ and fake $\tilde{x}$
7     |   Compute the discriminator loss, backpropagate and update $D_{\hat{\phi}}(x)$ using $\alpha_{\text{inner}}$
8     |   Compute the generator loss, backpropagate and update $G_{\hat{\theta}}(z)$ using $\alpha_{\text{inner}}$
9   **end**
10   **Meta-update generator**: $\theta \leftarrow \theta - \alpha_{\text{meta}}(\theta - \hat{\theta})$
11   **Meta-update discriminator**: $\phi \leftarrow \phi - \alpha_{\text{meta}}(\phi - \hat{\phi})$

---

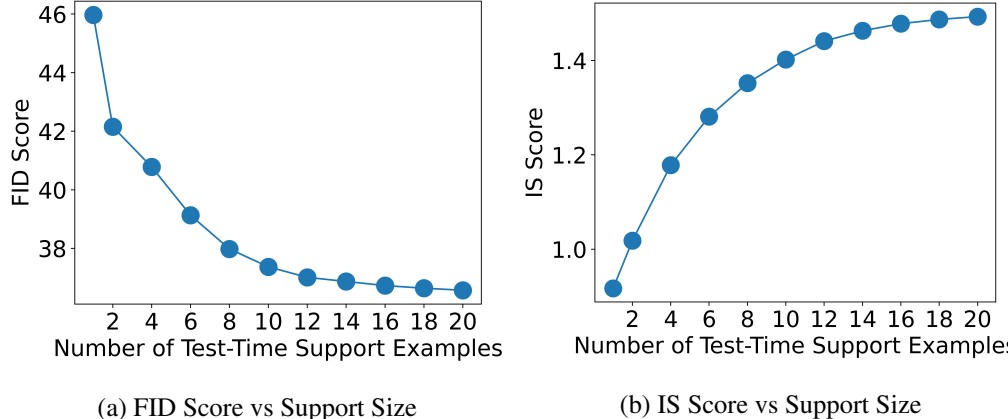

(a) FID Score vs Support Size          (b) IS Score vs Support Size

Figure 3: Plotting the FID and IS scores of the generated nines at test time with varying number of support examples (1 to 20) available for the model to learn from. Intriguingly, even $k_s = 1$, i.e. one-shot learning, exhibits a reasonable FID score of around 46. Each successive increase in the support set size results in a smaller improvement in both FID and IS, indicating diminishing returns.

## B  ADDITIONAL EXPERIMENTS

### B.1  META-TRAINING TIME

As seen in Table 1 (for MNIST) and Table 2 (for SVHN), both MAML and Reptile took substantially shorter ($3\times$ to $4\times$) to meta-train to convergence than the baseline, and ultimately achieved the same, if not better, performance. We hypothesize that meta-training enables the model to most efficiently learn in this setting: because each meta-training task presents a novel digit with a high amount of information waiting to be learnt, and because stochastic updates are made after each meta-episode (fine-tuning on a single task), the model rapidly learns how to generate novel digits given only a few support examples. On the other hand, since baseline training is only done on batches of all nines, the model cannot learn useful features from the other digits; concretely, the amount of information available to be learnt in each batch is less. Thus, the model takes longer to learn the distribution and converge to the indicated level of performance. Additionally, as expected, TransGAN takes longer to meta-train compared to the equivalent Res-SNGAN run, given that it is a larger/more complex model.

Further, we observe that MAML and Reptile achieve roughly the same performance (qualitatively and quantitatively) on MNIST & SVHN with Res-SNGAN & TransGAN, despite Reptile taking roughly 30-40% less time to meta-train. This reinforces the findings of Nichol et al. (2018) for few-shot image generation: because of Reptile's first-order simplification, it is able to train with greater stability and efficiency, reducing the total training time needed to converge without sacrificing on performance.

### B.2  IMPACT OF TEST-TIME SUPPORT SIZE

We evaluate the impact of the size of the test-time support set ($k_s$), varying from 1 to 20, on the quantitative performance of Reptile. Recall that this is the number of examples of '9' available for the meta-trained model to learn from during test-time fine-tuning. We use the Res-SNGAN model meta-trained on MNIST. Figure 3 plots the FID and IS scores of the generated nines as a function of the number of test-time support examples $k_s$. Intriguingly, even $k_s = 1$, i.e. one-shot learning, exhibits a reasonable FID score of around 46, indicating that even with a single training sample, the meta-trained model is able to successfully adapt to the test distribution and generate somewhat-photorealisic images of nines. For each additional example added to the support set, the corresponding improvement in both FID and IS score becomes smaller, indicating diminishing returns of larger support sizes. In particular, a support of $k_s = 20$ exhibits an FID of only about $0.6$ better than $k_s = 10$, while increasing from $k_s = 2$ to $k_s = 10$ results in an improvement of nearly 5 points.

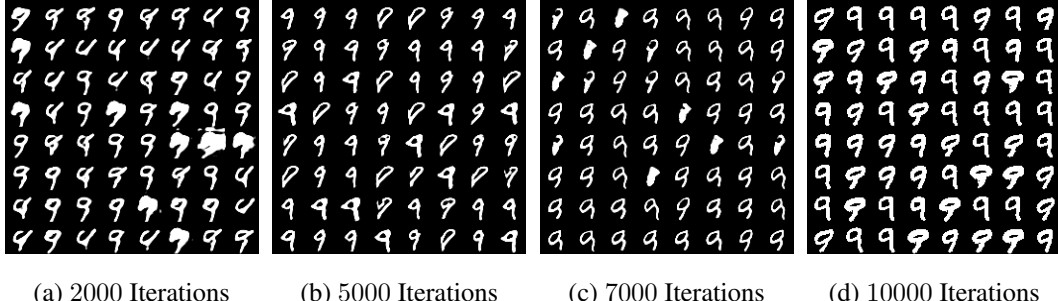

| (a) 2000 Iterations | (b) 5000 Iterations | (c) 7000 Iterations | (d) 10000 Iterations |

Figure 4: Visualizing the progression of Reptile training on MNIST with Res-SNGAN. Even after 2000 meta-training iterations, the generated nines look somewhat photorealistic. As meta-training continues, the generated nines (fine-tuned on 10-point support) keep improving visually, as expected.

### B.3 VISUALIZING REPTILE META-TRAINING

In Figure 4, we visualize the progression of meta-training with Reptile by examining the generated nines after 2000, 5000, 7000, and 10000 meta-training iterations (i.e. number of meta-tasks trained on). We train with Res-SNGAN on MNIST. As can be seen, even after 2000 meta-training iterations, the generated nines after fine-tuning to the support set are relatively photorealistic. Of course, training for more meta-iterations results in better quality results, with the final model trained for 10000 iterations exhibiting the best generated nines. All fine-tuned on 10-element support sets at test time.

### B.4 CONFIRMING BASELINE CANNOT ADAPT

In order to assert that our meta-training algorithms truly enable fast adaptation to the limited support set at test time, we must confirm that a vanilla-trained model cannot also adapt to a 10-element support and generate high-quality images. We vanilla-train a Res-SNGAN on the entire MNIST training set *excluding* all the images of '9's (i.e. on all the images of digits '0' through '8'). We train for 50000 iterations, keeping the other hyperparameters same as that of baseline training. We then take the fully-trained model, fine-tune it to a 10-element support set of '9's, and evaluate the generated results. As expected, this model often doesn't even generate the number '9': it exhibits an FID, KID, and IS of 245.87, 0.1931, and 1.329 respectively when evaluated against the set of images of '9' from the test set. The FID and KID scores in particular are nearly $10\times$ worse than the scores for MAML and Reptile, showcasing the power of meta-training in achieving few-shot adaptation.

## C RELATED WORK

**Generative Adversarial Networks** Modern GAN architectures typically consist of separate generator and discriminator models parametrized by deep networks (CNNs, Transformers, etc.) (Goodfellow et al., 2014). During training, the generator is trained to generate images $x_{\text{fake}}$ from the latent space $z$, such that the discriminator cannot tell apart $x_{\text{fake}}$ from the training data $x_{\text{real}}$. Concretely, GANs learn via the following mini-max optimization framework, where $m$ denotes the training batch size:

$$\min_\theta \max_\phi \mathcal{L}_{\text{GAN}} = \frac{1}{m} \sum_{i=1}^{m} \log D_\phi(x^{(i)}) + \log(1 - D_\phi(G_\theta(z^{(i)}))) \tag{1}$$

The first architecture to utilize a CNN to parametrize the generator $G_\theta$ and discriminator $D_\phi$ was the **DCGAN** (Radford et al., 2016). Subsequently, **SN-GAN** (Miyato et al., 2018) introduced a novel weight initialization scheme, spectral normalization, to avoid mode collapse, improving stability and performance. **Res-SNGAN** improves upon this architecture by parametrizing the generator and discriminator with ResNet convolutional networks, highly successful in other computer vision domains. Recently, **StyleGAN** (Karras et al., 2019) and **StyleGAN-V2** (Karras et al., 2020) have become the state of the art CNN-parametrized GANs, introducing multi-level mappings from the latent space into the generator to more explicitly control the quality and style of the generated image.

**Transformer-Based GANs**  Given the success of Transformers across machine learning (Vaswani et al., 2017; Dosovitskiy et al., 2021), several prior works have attempted to parametrize GANs with Transformers. **ViTGAN** (Lee et al., 2021) utilizes the large-scale Vision Transformer (ViT) network (Dosovitskiy et al., 2021), which has achieved state-of-the-art results on several image classification benchmarks, to parametrize the generator and discriminator. Developing novel normalization techniques for the model, the authors found that ViTGAN successfully outperformed StyleGAN-V2 on CIFAR-10, CelebA, and LSUN image generation benchmarks. Similarly, **TransGAN** (Jiang et al., 2021) developed convolution-free Transformer architectures for the generator and discriminator, along with a novel grid self-attention layer to alleviate memory bottlenecks and enable further scale-up of the model. The authors also introduce several techniques to improve training stability, such as improved normalization and data augmentation. TransGAN was also shown to outperform StyleGAN-V2 on CIFAR-10 and CelebA, both qualitatively and quantitatively (using the FID, KID, and IS evaluation metrics). It is reasoned that Transformers are few-shot learners: when pre-trained on a massive dataset, the network can easily adapt to a new task (and achieve state-of-the-art results) with just a few fine-tuning steps relative to the convolutional baseline (Dosovitskiy et al., 2021).

**Few-Shot Meta-Learning**  The challenge of meta-learning for few-shot image classification has been richly investigated. Model-Agnostic Meta-Learning (MAML) (Finn et al., 2017) is a simple yet powerful algorithm for meta-training an image classifier for fast adaptation on a sparse (few-shot) target task. Concretely, the algorithm meta-trains the model to learn a weight initialization (outer loop) such that, given a target task with few datapoints, the model is able to rapidly learn the decision boundaries in the task with just a few gradient steps (inner loop). Yet, MAML often suffers from instability and sensitivity to initialization schemes, in part because it requires the loss to be backpropagated through the entire inner loop at train time. To make the algorithm simpler, Reptile (Nichol et al., 2018) imposes a first-order approximation on the MAML gradient, no longer requiring a full back-propagation through the inner loop. As a result, Reptile is much more stable, less sensitive to initialization, and trains significantly faster, with only a negligible loss in performance.

**Few-Shot Image Generation**  While the aforementioned GAN architectures are highly effective at learning the training distribution, they require extremely large and diverse datasets in order to do so (Ojha et al., 2021). As such, several works have tried to develop strategies for training GANs in the few-shot setting, where only limited data is available to learn from. Ojha et al. (2021) studied the challenge of adapting GANs pre-trained on a large source dataset to a target dataset with less than 10 training examples. Devising a novel cross-domain consistency metric to preserve the relative similarities and differences of images in the source when transformed onto the target, the authors showcased significant improvements of their model in both photorealistic and non-photorealistic domains. To further avoid overfitting, the authors utilized an anchoring technique to segregate regions in the latent space with respect to their stylistic effects. Concurrently, Robb et al. (2020) proposed FSGAN for few-shot image generation. The technique performs a singular value decomposition of the pre-trained generator model weights, freezes the singular vectors, and only trains the singular values on the target dataset (with $< 100$ examples). Liang et al. (2020) also adapted the MAML and Reptile few-shot learning algorithms to the GAN setting, meta-training GANs for fast adaptation on music and image generation tasks. However, they only considered CNN-based GANs (not Transformers), only tested on MNIST, and, as we demonstrated earlier, used an incorrect strategy for meta-training. Several other works have also explored the challenge of few-shot image generation with GANs & other models (Wertheimer et al., 2020; Clouâtre et al., 2019; Fontanini et al., 2020; Hong et al., 2020).

## D  EXPERIMENTAL DETAILS

### D.1  GAN ARCHITECTURES

**Res-SNGAN**  We use the same GAN architecture introduced by Miyato et al. (2018), with both the generator and discriminator parametrized by ResNets (same as used in their paper). We perform spectral normalization weight initialization to improve stability and prevent mode collapse (key contribution of the Res-SNGAN paper). We train these models from scratch on MNIST and SVHN.

**TransGAN**  As in Jiang et al. (2021), we use a Vision Transformer (ViT) network (Dosovitskiy et al., 2021) to parametrize the generator and discriminator, along with the same weight initialization

schemes used in the paper to improve stability. See Figure 2 in Jiang et al. (2021) for a full diagram of TransGAN's architecture. We utilize TransGAN pre-trained on the task of CIFAR-10 image generation, as Transformers/ViT have been shown to be computationally expensive to train from scratch and quite effective at fine-tuning to different tasks when pre-trained (Dosovitskiy et al., 2021).

Note: we scale all images (MNIST and SVHN inputs, generator outputs) to be $32 \times 32$ images. We utilize a 128-dimensional vector latent space for the generators of both Res-SNGAN and TransGAN.

## D.2 QUANTITATIVE EVALUATION

**Frechet Inception Distance (FID)**    We compute FID scores for a set of synthetic images following the standard process in the GAN literature, embedding the real & synthetic images using a pre-trained InceptionV3 model and calculating the Frechet distance. Lower FID scores mean better results.

**Kernel Inception Distance (KID)**    We compute KID scores using the standard process in the GAN literature as well, embedding images using the pre-trained InceptionV3 and calculating the maximum mean distance. Lower KID scores mean better results. Both FID and KID measure how closely the distribution of synthetic/generated images matches the distribution of real images for a given task.

**Inception Score (IS)**    We again follow the standard process for computing IS scores in the GAN literature, using the pre-trained InceptionV3 to compute classification scores and entropy scores. IS measures both the quality and diversity of synthetic images, and higher scores are considered better.

