# OpenReview forum: "Meta-GAN for Few-Shot Image Generation"
_ICLR.cc/2022/Workshop/DGM4HSD — ICLR 2022 DGM4HSD workshop Poster_

### Official Review · Reviewer_Z9D3 · 2022-03-21
**Well-written paper, but the usefulness to the community is not very clear to me**

**Rating:** 4
**Confidence:** 3

**Review:**

The authors study few-shot image generation using the MNIST and SVHN datasets by meta-learning CNN or Transformer-based GANs on single-digit tasks, consisting of the digits ‘0’-’8’, and fine-tuning on a small number of examples of the single-digit task of modeling ‘9’. The paper is well-written and it consists of thorough experiments. Some questions and concerns remain.

I am not sure about the novelty of the paper. Apart from training a Transformer-based GAN, what is the technical novelty of the paper? One could claim that the observation of the faster convergence of meta-pret-training is novel, but the training curves are missing from the paper. I do not seem to find a detailed discussion about this phenomenon in the paper.

Natural baselines are missing: 1) joint (multi-task) pre-training; 2) Transformers pre-trained with less data (e.g. randomly-initialized Transformer). Could the authors compare against these baselines?

Related work: An important few-shot image generation related work is missing from the discussion: https://arxiv.org/abs/2112.11929. It is not clear how the paper is distinguished from other works on few-shot image generation, e.g. the last sentence of the “Few-Shot Image Generation” paragraph in the appendix. I suggest the authors provide a convincing clarification.

---

### Official Review · Reviewer_UfpN · 2022-03-23
**Advances few-shot image generation using meta learning but lacks a rigorous experimental setup.**

**Rating:** 6
**Confidence:** 4

**Review:**

This work improves on few-shot image generative by integrating meta-learning algorithms with generative adversarial networks (GANs). It uses two meta-learning approaches, namely MAML and Reptile meta-learning. It demonstrates that GANs pre-trained with meta-learning objectives generate high fidelity images when finetuned on a very small number of images from the domain in downstream tasks.

As shown in fig.1 proposed approach successfully generates high fidelity images. However, the coverage or diversity of these images is relatively poor. This is expected since only a very small number of images are used in finetuning. Nevertheless, it would be a strong addition to the paper to include metrics (recall from precision-recall [1]) that captures the diversity. It will validate whether images in fig. 1(b, c) are suffering from model collapse, as most of them are very similar to each other.

Another suggestion is to consider a setup whether downstream and upstream tasks don't share the same domain. E.g., when pre-training on digits 0-8 and fine-tuning on digit 9, the models benefit a lot by the fact that all digits are from the same datasets and have similar image semantics. A more challenging setup would be pre-training on colored images and finetuning to grayscale ones, or vice-versa.

[1] Kynkäänniemi, Tuomas, et al. "Improved precision and recall metric for assessing generative models." Advances in Neural Information Processing Systems 32 (2019).

---

### Official Review · Reviewer_dCh6 · 2022-03-23
**Clear, well-executed empirical paper on meta-learning for image GANs. Possible mismatch with the workshop's theme**

**Rating:** 7
**Confidence:** 4

**Review:**

**Overview**: The paper investigates few-shot image generating for modern GANs. Specifically, they explore how two popular meta-learning algorithms (MAML and Reptile) perform in making TransGAN and ResGAN generate realistic samples from training (i.e., fine-tuning after meta-learning) on only ten samples. They experiment with MNIST and SVHN and show that meta-learned GANs adapted on ten samples are slightly inferior to GANs trained from scratch on ~500x more examples (in the case of MNIST). Note that total data for meta-training is ~5x the data for training from scratch. As expected, fine-tuning a meta-learned model takes ~3-4x less time than training from scratch.

Given that the workshop's organizers believe the paper's topic (image generation) is appropriate, I suggest that the paper be accepted.

My rating of 7 is in the context of the standard for a workshop publication (and not a full conference publication). The paper can become a full conference paper by expanding the investigation to more datasets and adding more complicated scenarios (e.g., meta-train on CIFAR and adapt on MNIST).

**Strengths**
- Useful empirical contribution in showing the potential of meta-learning for modern image GANs.
- Very clear and well-written paper. Easy to read. Nice appendix.
- Neat investigation. I also believe your approach on training on only one of 0-8 is the better approach because - the test distribution 9 is kind of "unimodal" because it contains one image type.

**Weaknesses**
- Possible mismatch with the workshop's theme. The paper investigates image generation, while the workshop aims to explore generative models useful in other highly-structured domains.
- My concern is that the meta-train and meta-test distributions are too similar. For example, if one trains on 3, 6 or 8, it's straightforward to learn to generate a 9. It would be great to see experiments train the meta-training is on something more out of distribution. For example, what results would you expect if you meta-train on FashionMNIST and test on MNIST?
- Little novelty. However, the empirical confirmation is useful for the community.

**Minor weaknesses**
- You say, "Transformers are simply better at image modelling than CNNs". There are recent papers that show their performance on vision is on par. Please expand and provide arguments.
- You say "convergence takes substantially shorter (3× to 4×) than baseline training". Why is it surprising? The training objective of MAML is to adapt quickly to the test distribution, so I would expect MAML-trained GANs to train quicker than the baseline.
- Consider putting KID as the last column in the tables. All methods perform essentially identical.

---

### Decision · Program_Chairs · 2022-03-27

Accept (Poster)